# Decrease in Soil Functionalities and Herbs’ Diversity, but Not That of Arbuscular Mycorrhizal Fungi, Linked to Short Fire Interval in Semi-Arid Oak Forest Ecosystem, West Iran

**DOI:** 10.3390/plants12051112

**Published:** 2023-03-01

**Authors:** Javad Mirzaei, Mehdi Heydari, Reza Omidipour, Nahid Jafarian, Christopher Carcaillet

**Affiliations:** 1Department of Forest Science, Faculty of Agriculture, Ilam University, Ilam 69315-516, Iran; 2Department of Rangeland and Watershed Management, Faculties of Natural Resources and Earth Sciences, Shahrekord University, Shahrekord 8818634141, Iran; 3Ecole Pratique des Hautes Etudes (EPHE), Paris Sciences & Lettres Université (PSL), F-75014 Paris, France; 4Univ Lyon, Université Claude Bernard Lyon 1, CNRS, ENTPE (UMR 5023 LEHNA), F-69622 Villeurbanne, France; 5Department of Plant Sciences, University of Cambridge, Cambridge CB2 3EA, UK

**Keywords:** AMF, enzyme activity, forest, disturbance, biodiversity, soil, functional ecology

## Abstract

The semi-arid forest ecosystems of western Iran dominated by *Quercus brantii* are often disturbed by wildfires. Here, we assessed the effects of short fire intervals on the soil properties and community diversity of herbaceous plants and arbuscular mycorrhizal fungi (AMF), as well as the interactions between these ecosystem features. Plots burned once or twice within 10 years were compared to unburned plots over a long time period (control sites). Soil physical properties were not affected by the short fire interval, except bulk density, which increased. Soil geochemical and biological properties were affected by the fires. Soil organic matter and nitrogen concentrations were depleted by two fires. Short intervals impaired microbial respiration, microbial biomass carbon, substrate-induced respiration, and urease enzyme activity. The successive fires affected the AMF’s Shannon diversity. The diversity of the herb community increased after one fire and dropped after two, indicating that the whole community structure was altered. Two fires had greater direct than indirect effects on plant and fungal diversity, as well as soil properties. Short-interval fires depleted soil functional properties and reduced herb diversity. With short-interval fires probably fostered by anthropogenic climate change, the functionalities of this semi-arid oak forest could collapse, necessitating fire mitigation.

## 1. Introduction

Wildfires are a major disturbance in many biomes around the world, including the tropical, Mediterranean, temperate, and boreal, but there are exacerbated in the arid and semi-arid regions, e.g., [1,2,3,4]. Drought, grazing, and wildfires jointly control the expansion of savanna/steppe ecosystems and their potential aridification [2,5]. Wildfires can drive long-term changes in the ecosystem structure and functioning by affecting wildlife, surface runoff, air quality, the microclimate, soil, etc., depending on the time since the last fire and the frequency, intensity, size or type of fire [6,7,8]. 

Plant traits covary with wildfires [9]. Fire can change their physiology in terms of rate or efficiency, demography or gamma diversity, which thus affect biotic interactions and, ultimately, community dynamics and productivity [7,10,11,12]. Notably, plant dynamics can be controlled by fire products, i.e., ash, charcoal, and smoke [13,14].

Wildfire affects soil microorganisms [15,16,17], including fungi [6,18,19], due to changes in soil properties or partner plants [20,21]. Soil biological properties would be more sensitive to disturbances than physical and chemical ones, and their response should be faster and broader [22]. Because microorganisms control biochemistry [16], they are key proxies of ecosystem quality [23,24], which is useful for sustainable management that requires the needs assessment of factors driving the variation in soil bio-properties [25]. Therefore, post-fire changes in soil biological features could be a warning signal for ecosystem stress and eventually for steady state change [26]. Among biological properties, enzyme activities, such as urease and soil respiration, are fire-sensitive [16,27,28].

At the landscape scale, soils interact with wildfires, controlling the biomass and diversity of plants and microorganisms [29,30]. Wildfires change the composition and load of soil organic matter [15], which could lead to variations in the physical and chemical soil properties [31,32,33]. The decrease in the soil’s organic carbon load depends on the duration and intensity of wildfires [34].

The semi-arid forest is the most widespread ecosystem in the Zagros Mountains in western Iran [35]. The Zagros forests are fire-prone due to summer droughts and the dense understory canopy, as well as the presence of farmers and other land users. The future climate is expected to increase the fire risk in this area unless net primary productivity decreases [36]. Although there are few studies in the Zagros forests reporting the effect of a single fire on soil physicochemical properties [37,38], the cumulative ecological effects of a short fire interval represent a gap of knowledge in this ecosystem, but their long-term effects could be crucial for soil processes and biodiversity [39,40,41].

The present study aims to analyze the consequences of a short fire interval on soil properties in association with plant and fungal diversity in the semi-arid oak forest ecosystem of the Zagros Mountains. We hypothesize that after a short fire interval, soil biological properties (organic matter, respiration, enzymatic activity) should be altered, triggering a decrease in the alpha diversity of arbuscular mycorrhizal fungi (AMF) or vascular plant communities, resulting in a collapse in functionalities. Due to biomass consumption, cations are released after fire [42], and oxides, hydroxides, and carbonates of sodium or potassium should increase [10,43]. An increase in salinity is therefore expected with a short fire interval.

## 2. Materials and Methods

### 2.1. Study Site and Fire History

The study area (46°26′–46°28′ E, 33°36′–33°37′ N) is in west Iran, near Ilam City (Figure 1). This region is hilly to mountainous and presents gentle to steep slopes. The climate is semi-arid with a mean annual temperature of 16.7 ± 0.7 °C and a mean annual precipitation of 591 ± 173 mm. Most of the precipitation occurs from October to March. The altitudes of the study area range between 1742 and 1972 m a.s.l., with slope degrees between 20 and 40%. Soils are sandy silt (ca. 60% sand, 20% silt and 20% clay), alkaline based on limestone bedrocks (pH > 7) and shallow (depth < 50 cm). Zagros forests are dominated by Brant’s oak (*Quercus brantii* Lindl), which is also widely distributed in Turkey, Iraq, Syria and Lebanon [44]. The species composition of the forest includes notably *Pistacia atlantica* Desf., *Acer monspessulanum* L., *Cerasus microcarpa* C. A. Mey. and *Daphne mucronata* Royle. 

**Figure 1 plants-12-01112-f001:**
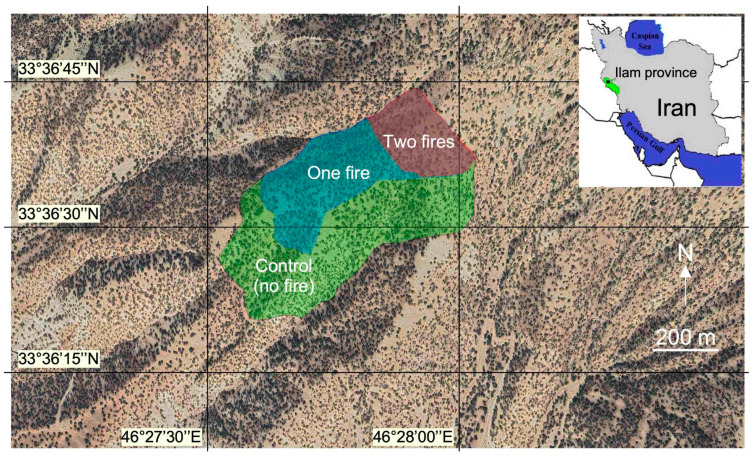
Aerial view (Image ©2022 Airbus) of the studied area in the Ilam Province, western Iran; fire polygons of different treatments (two fires; one fire; control site where no fire occurred for 10 years at least). The image illustrates well the light tree cover (dark pixels) and the visible ground (light pixels) typical of steppe forests and hilly topography.

Information about the perimeters and dates of wildfires was obtained from the Ilam Natural Resources Office. We selected 3 different sites (Figure 1) that were burned with a high fire intensity once or twice during the last 10 years [45]. The area that was burned once in 2017 is hereafter coded “OF,” those that were burned twice in 2017 and 2019 are coded “TF”, and the control area that did not burn was coded “Con”. Both had the same physiography, and we assumed that all factors were equivalent between these fires, except the date (Figure 1).

### 2.2. Data Collection for Physics and Chemistry of Soil

Sampling was conducted in spring 2020 based on seven 20 × 20 m (400 m^2^) plots for each of the three treatments (OF, TF, and Con), randomly established (in total, 21 plots). Four 1.5 × 1.5 m (2.25 m^2^) quadrats were established in each plot to assess the vascular herbaceous plant community (grass and forbs), with 84 quadrats in total. Three soil samples per plot were randomly collected at 0–20 cm depth. The three soil samples were mixed together to obtain two sub-samples for soil analyses and one for the arbuscular mycorrhizal fungi (AMF) analysis; one for soil analyses was stored at +4 °C in the refrigerator before measuring microbial and biochemical properties (microbial biomass, basal respiration and urease enzyme), and the second was air-dried and used for the evaluation of physical and chemical properties, such as texture (clay, silts, sands), bulk density, pH, electrical conductivity (EC, proxy for salinity), macro-nutrients (N and P), organic matter, and total neutralizing value (TNV). The soil analysis methods are summarized in Table 1.

### 2.3. Diversity of Vascular Plants and Arbuscular Mycorrhizal Fungi

For the arbuscular mycorrhizal fungi (AMF) analysis, spores were extracted from the soil following the wet sieving method [57]. The microscopic identification of AMF spores (100×) was based on their morphological characteristics following the taxonomic criteria [58], supported by the reference website of the International Collection of Vesicular AMF (INVAM, www.invam.caf.wvu.edu). After the identification and density assessment of herbs and forbs [59] for vascular plants, and AMF taxa, their biodiversity was calculated based on diversity indices whose formulas are presented in Table 2, as well as their ranges and properties. 

### 2.4. Statistical Analysis

Prior to analysis, the normality of means and homogeneity of variances were assessed using Shapiro–Wilk and Leven′s tests, respectively. A one-way analysis of variance (ANOVA) was used, followed by Dunca′s multiple range tests (in case of significant effect) to compare the effects of fire occurrences on the soil measurements and diversity indices of plants or AMF communities. To explore the statistical interactions and antagonisms among factors in each of the 3 areas, principal component analyses (PCA) were used. This analysis was carried out based on ANOVA outputs, in which we only included significant variables.

To detect the direct and indirect effects of fire frequency on soil physical, chemical, and biological features and on diversity indices, a structural equation modeling (SEM) was implemented. A multi-path diagram was first designed (Appendix A) for the direct and indirect effects of fire frequency. Then, a confirmatory SEM procedure was used to assess the statistical significance of the paths. To simplify the final model, the best variables in each group were per-selected prior to SEM analysis based on non-parametric Spearman tests (Appendix A). To assess the strength of SEM models, several statistical indicators were calculated, including the Chi-square test (*χ*^2^ test, *p*-value > 0.05), goodness-of-fit index (GFI > 0.95) and comparative fit index (CFI > 0.95). All statistical analyses were carried out under the R environment, version 4.1.0 (R Core Team, 2021), using the “FactoMineR,” “factoextra,” and “lavaan” packages.

**Figure 2 plants-12-01112-f002:**
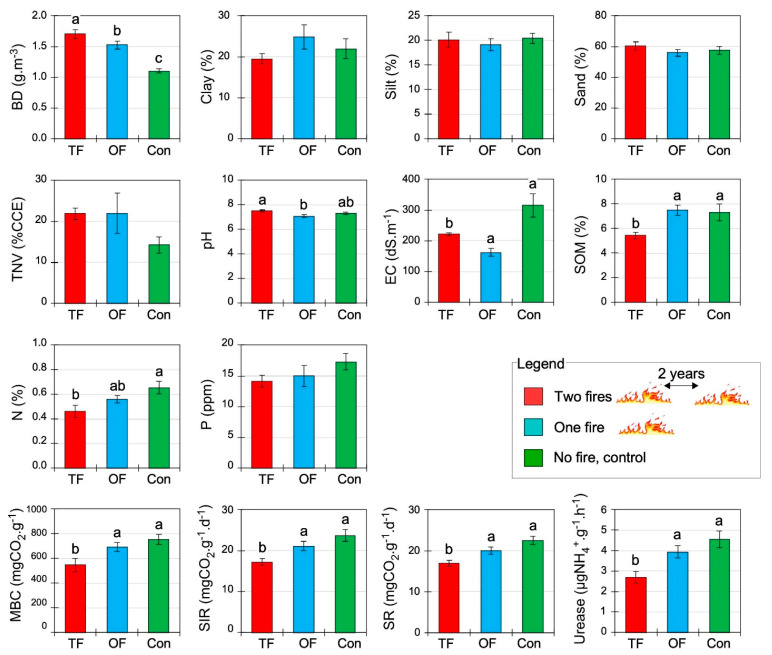
Soil physical, chemical, and biological properties (mean ± SE) under different fire treatments: two fires (TF), one fire (OF), and no fire occurred or control site (Con). BD: bulk density; TNV: total neutralizing value; EC: electrical conductivity; SOM: soil organic matter; pH: soil acidity; N: nitrogen; P: phosphorus; MBC: microbial biomass carbon; SR: soil respiration; SIR: substrate-induced respiration; Urease: soil urease enzymatic activity. Letters indicate significant differences at *p* < 0.05 inferred from ANOVA followed by Duncan’s multiple range tests.

## 3. Results

### 3.1. Effect of Fire Treatment on Soil Properties

All soil biological properties, most chemical properties, and one physical property were significantly affected by the fire number (*p* < 0.05; Table 3). Among the physical properties, bulk density (BD) increased from around 43% and 54% in OF (one fire) and TF (two fires), respectively, in comparison to the control area, whereas the granulometry (sand, silt, and clay) was not affected (Table 3, Figure 2). Among the chemical soil properties, electrical conductivity (EC) was considerably lower in both burned areas compared to the control. Regarding SOM and N, only TF presents lower significant values. The highest and lowest levels of pH were observed in the case of TF (7.5 ± 0.1) and OF (7.1 ± 0.1), and the control site was intermediate (7.3 ± 0.1). However, there was no significant difference between the phosphorus (P) concentration and lime content (TNV) in the three treatments (Table 3, Figure 2). The results showed that all soil biological properties decreased in response to TF (Table 3, Figure 2). Indeed, TF treatment decreased substrate-induced respiration (SR), basal soil respiration (SIR), microbial biomass carbon (MBC), and urease by approximately 33, 38, and 70% compared to the control site, respectively (Figure 2). The more fires, the lower the soil biological activity and properties.

### 3.2. Effects of Fire Treatment on Community Diversity

Fire frequency had an effect on the herb community in terms of rarefied richness (*R*, *p* < 0.01), the Gini-Simpson index (*D*, *p* < 0.01), and the Shannon index (*H*′, *p* < 0.01; Table 3, Figure 3). The highest and the lowest values were found in OF and TF, respectively, and the reference area was intermediate. Because *R*, *D* and *H*′ have increased with OF compared to the control, this indicates that the species number has increased (*R*) and that their relative abundances per species tended to balance out (*D*, *H*′). However, despite the increase in *D* and *H*′, the evenness (*E*) of the community did not change (Table 3, Figure 3).

Regarding the biodiversity of arbuscular mycorrhizal fungi (AMF), only the Shannon index (*p* = 0.01) was significantly affected by fire occurrences with greater diversity in both cases of TF (2.06 ± 0.04) and OF (2.05 ± 0.03). However, other AMF diversity indices (*R*, *D*, and *E*) were not significantly changed by fires (Table 3, Figure 3), although their values in burned areas were slightly greater than in the control. The fact that only the *H*′ slightly increased for AMF indicates that the change in fungal diversity mainly concerned non-abundant taxa but with no change in the species number (*R*).

**Figure 3 plants-12-01112-f003:**
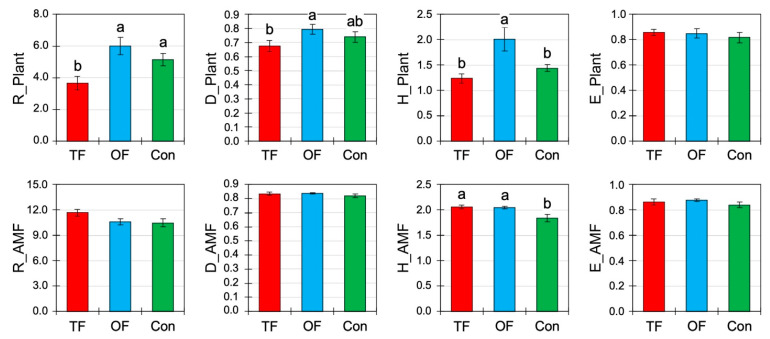
Herb and arbuscular mycorrhizal fungi community diversity (mean ± SE) under different fire treatments: two fires (TF), one fire (OF), and no fire occurred or control site (Con). *R*: Menhinick richness; *D*: Gini-Simpson index; *H′*: Shannon index; *E*: Pielou’s evenness index. Letters indicate significant differences at *p* < 0.05 inferred from ANOVA followed by Duncan’s multiple range tests.

### 3.3. Soil Properties and Biodiversity under Different Fire Treatments

A holistic analysis of soil and community parameters was conducted thanks to a principal component analysis (PCA). The PCA outputs showed that around 65% of the overall variance was explained by axes 1 and 2 together (46 and 19%, respectively), underscoring that the whole variance of the dataset was well captured by only two axes (Figure 4, Appendix A). We observed a clear distinction between the control area and the two-fire treatment (TF) along axis 1 (PC1), which was driven by soil properties including SR (r = 0.94, *p* < 0.001), N (r = 0.94, *p* < 0.001), urease (r = 0.93, *p* < 0.001), SIR (r = 0.91, *p* < 0.001) and MBC (r = 0.89, *p* < 0.001; Appendix A), which is a soil functionality gradient. PC2, which is a biodiversity gradient, is driven by and positively correlated to plant diversity indices, including Shannon (r = 0.80, *p* < 0.001), Gini–Simpson (r = 0.67, *p* < 0.001), and evenness (r = 0.66, *p* < 0.001; Appendix A). One-fire plots are separated from both the control and the two-fire areas along the positive PC2.

### 3.4. Direct and Indirect Effects of Fire on Biological Diversity and Soil Propperties

The pre-selection analysis showed that soil BD, N, SIR, and plant richness and fungal Shannon diversity indices were the best variables in terms of physical, chemical, and biological properties and biodiversity (Appendix A). Overall, the structural equation modeling (SEM) evaluating the direct and indirect effects of different fire treatments on each soil and community parameter yielded an adequate data fit (RMSEA = 0.00; *χ*^2^ = 8.49, *p* = 0.486, CFI = 1, GFI = 0.987; Figure 5). SEM-based standardized total effects (sum of significant direct and indirect effects) revealed that fire occurrences significantly and positively altered the soil physical properties represented by BD (β = 0.85, *p* < 0.001) and AFM Shannon diversity (β = 0.39, *p* < 0.05; Figure 5, Appendix A). Conversely, fire occurrences negatively affected the soil chemical properties represented by the total N concentration (β = −0.46, *p* < 0.05), soil biological properties represented by SIR (β =−0.68, *p* < 0.001) and plant diversity represented by species richness (R, β = −0.38, *p* < 0.05; Appendix A). Finally, fire occurrence indirectly alters fungal diversity (β = 0.194 and *p* < 0.05) through the mediatory role of chemical properties (Figure 5, Appendix A).

**Figure 4 plants-12-01112-f004:**
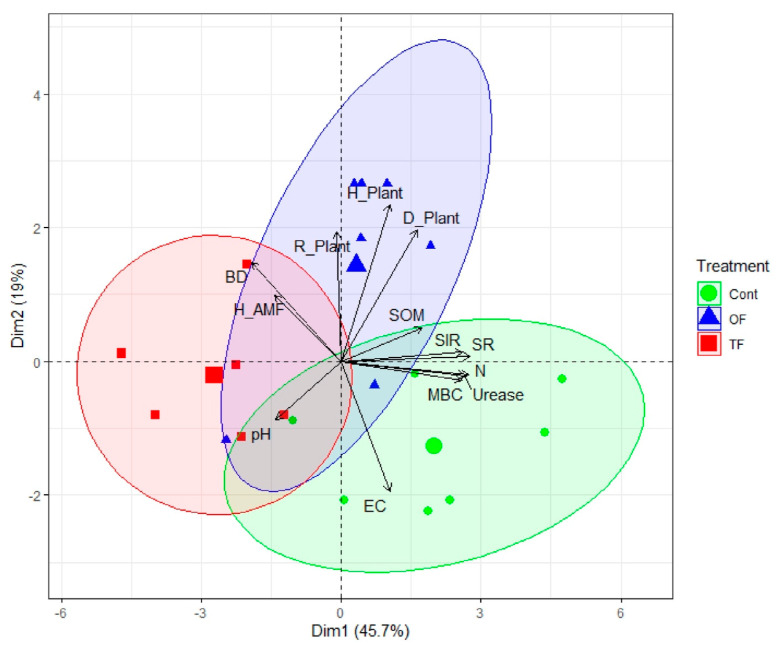
The fire treatment predicts the soil functionalities and biodiversity. Biplot of principal component analysis (PCA) based on soil physical, chemical, and biological properties, and fungal and plant community diversity indices, under different fire treatments: two fires (TF), one fire (OF), and control site (Con). BD: bulk density; EC: electrical conductivity; pH: soil acidity; SOM: organic matter; N: total nitrogen; SR: soil respiration; SIR: substrate-induced respiration; MBC: microbial biomass carbon; Urease: urease activity; Biodiversity indices: richness (R), Gini-Simpson (D), Shannon (H) for herbs (Plant) of fungi (AMF). The centroid of plots under TF, OF, or Con treatment is indicated by a greater shape (red square: TF; blue triangular: OF; green circle Con). Ellipses denote the perimeter of plots per treatment.

## 4. Discussion

The present study supports our working hypothesis, assuming that a fire in a short interval should alter the soil’s biological properties, triggering a decrease in plant diversity and, thus, a collapse in the functionalities of the semi-arid oak forest ecosystem of the Zagros Mountains. However, the effects on the diversity of arbuscular mycorrhizal fungi were not so obvious and were possibly delayed. Finally, the hypothesized increase in salinity (EC) due to biomass consumption was not reported, refuting the original hypothesis regarding salinity.

**Figure 5 plants-12-01112-f005:**
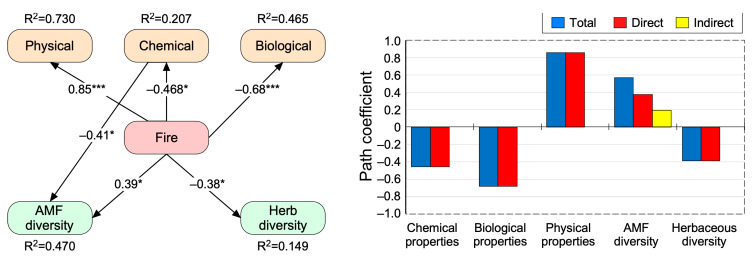
Direct and indirect effects of the fire treatment. The best fit structural equation modeling (SEM) on the best variables in different groups of soil physical, chemical, and biological and plant and fungal species diversity indices. The values adjacent to lines are standardized path coefficients. * and *** indicate the significant paths at 0.05 and 0.001, respectively.

### 4.1. Short Fire Intervals Increased Bulk Density and Decreased Electrical Conductivity

Bulk density was the sole physical attribute affected by the short fire interval (Table 3), increasing with two fires (Figure 2). This phenomenon has been frequently reported [32,33,60,61], although other studies have reported a decrease after a fire [62,63]. These discrepancies can be related to different environmental fire contexts, notably the fire intensity, soil type and, eventually, climate or vegetation types. Fire increases the bulk density by collapsing aggregates and clogging voids with ashes, thus causing the shrinkage of soil macropores and a reduction in porosity [27,32].

Regarding soil geochemistry, the analyses show that two fires slightly increase the soil pH and decrease soil electrical conductivity (Table 3, Figure 2), which is similar to other studies in semi-arid areas [64] or under different climatic contexts [65,66]. As a result of the burning of organic matter following a fire, cations are released [42], and oxides, hydroxides and carbonates of sodium and potassium are increased [43]. Therefore, a salinity increase was postulated following an analogous study [11], but this is not the case here, indicating that fire-salinity relationships are indirect and more complex. These differences could be explained by the important slope degree of the present study area (20 to 40%). Such steep slopes limit the burial of ashes due to runoff and, as a result, lead to a reduced EC. 

### 4.2. Depleted Biochemical Soil Properties

Two fires significantly decreased the concentration of soil organic matter (SOM) through pyrolysis [34]. Such an expected result in arid forests showing a decrease in SOM with fires supports similar outputs in shrub-dominated ecosystems [67] and is consistent with studies on Mediterranean forest soil [68]. Because SOM is a key soil property that supports soil nutrient availability and biological activity [69] and ecosystem functioning [70], the decrease in the SOM concentration with two fires at a short interval thus evidences a decrease in ecosystem functionalities.

Soil enzymatic activity, which is a proxy of microbial activity [71], decreased as the fire number increased (Figure 2). Enzymatic activity plays an important function in catalyzing the biotic reactions needed for SOM decomposition and nutrient recycling [71]. Here, due to fires, the understory plant species and litter were consumed and, as a result, the enzymatic activity was greatly reduced (Table 3, Figure 2), especially in the two-fire area, where the plant understory and SOM did not successfully recover due to the short time since the last fire. This observation matches a similar study showing that basal respiration and microbial activity greatly declined immediately after a fire in a semi-arid forest [44]. We thus concluded that the soil biological properties are the most sensitive properties to fire repetition, rather than soil physical and chemical ones. 

The reduction in SOM covaried with the decrease in the soil biological activity, such as enzymatic activity and microbial respiration (SR, MBC, and SIR; Figure 4), which supports the need for high SOM to secure soil biotic processes [72]. Further, the total nitrogen decreased significantly in the soil after one fire and mostly after two fires (Figure 2), which was expected [15,32,73,74]. Although the plants, as primary producers, are an important living nitrogen compartment in the ecosystem [75], the burning of soil biota and SOM reduces diazotrophic microorganisms, realizing the N-fixation from the atmosphere to the soil, or the ability of others to recycle nitrogen from SOM [17,76]. Burning thus causes substantial decreases in total soil nitrogen (Figure 2) by releasing nitrogen oxides into the atmosphere [77]. Finally, a wildfire reduces the overall activity of microorganisms by reducing moisture and soil organic carbon, an essential substrate for their activity [61,78]. 

Overall, the biochemical properties of the soil were depleted in the two-fire area, based on the decrease in the concentration of SOM, total N, and biotic activities (SR, MBC, SIR, urease), demonstrating the negative functions of a short wildfires interval.

### 4.3. Biodiversity Changes in Herbs and Arbuscular Mycorrhizal Fungi

Wildfires have a significant effect on the biodiversity of the plant understory of oak steppe forests (Figure 4). Herbaceous richness and diversity showed a positive hump-shaped pattern in which they were greater with a single fire than with two fires or no fire (Figure 3). Therefore, a single fire can increase the plant species’ biodiversity while fires at short intervals deplete it. It appears that this increase enhances slightly the dominant species (*D*) but also and mostly the rare species (*H*′; Figure 3). 

Overall, fires can increase plant diversity in dry forests [7,79], probably by reducing the tree competition for light and nutrients [80]. Other studies reported that a fire facilitates the establishment of species that were absent prior to the fire [81,82]. In addition, the fire-inferred heating stress increases seed germination by releasing seed dormancy [13,45,83]. Notably, smoke is a fire-inferred driver, independent of heat, that can release dormancy [84,85,86]. Hence, positive effects on seed germination and further on plant biodiversity by fire products such as ashes and smoke compounds have been reported [45,87]. However, with two fires in the arid oak forest of west Asia, this diversity decreases, probably in relation to soil biochemical properties, as reported in Figure 2 and Figure 4. This observation matches others showing that repeated fires reduce the plant diversity and the structure of plant communities [88,89,90] and, in some cases, can eliminate species for a long time [10]. This pattern is in line with the concept of the “intermediate disturbance hypothesis” [91]. Indeed, based on the dynamic equilibrium model [92,93], a moderate level of disturbance has positive effects on diversity.

Contrary to the plant community, wildfires did not significantly affect the richness, Simpson diversity, and evenness of AMF (Figure 3) despite assumptions based on meta-analysis outputs of fungal richness [18]. Only the Shannon index shows higher values compared to the reference area, which was not expected, although Shannon diversity appears indirectly influenced by fire through the mediatory role of soil chemistry (Figure 5). Such indirect relationships were already reported [18]. This discrepancy between AMF diversity indices based on changing Shannon and steady Gini-Simpson index reveals that fires enhanced rare fungi taxa and did not alter the abundance of the dominant ones. This output probably results from the fact that fires stress the reproduction of fungal spores, especially since our sampling was based (i) on spores’ identifications while environmental DNA provides a more complete view of fungal community [94], and (ii) the sampling occurred exactly one year after the fire. 

Our observation is consistent with Sun et al. [95], who found that 2 years after fires, the impact of burning was limited and that more time was needed to detect consequences on AMF diversity, contrary to the global pattern that indicates soil fungi are adversely affected by fire and that the resilience would be rapidly occurring in few years [18]. Su et al. [96] focused on a period of three decades after a fire to measure its effects. Since the method and the strategy to assess fungal communities is crucial [19], the present result should be considered with caution. However, we can predict that the biodiversity of AMF is expected to decrease in the near future when a fire has occurred twice in a short interval because (i) AMF has a mutualistic relationship with herbs, whose biodiversity decreased immediately, (ii) AMF recycles the SOM that has dropped, and (iii) because the soil biological properties were depleted. Therefore, it can be concluded that plant diversity is more immediately sensitive to fire occurrence and repetition than arbuscular mycorrhizal fungal diversity, which seems more delayed. 

## 5. Conclusions

In the semi-arid oak forest of western Iran, fires at short intervals impacted the soil biochemical properties and plant diversity, while the arbuscular fungal diversity was less sensitive. Contrary to the theory suggesting that soil fertility increases after wildfire thanks to nutrients’ release, we observed that soil fertility dropped. This observation is likely due to the consumption of soil organic matter and biomass and possibly due to autumn rainfalls, which export residual matters thanks to the slope degrees. Further, fires severely stressed the microbial biomass and the enzymatic activity, especially when fires occurred several times. The more fires, the lower the soil’s functional properties in the context of the semi-arid oak forests of the Zagros Mountains. Based on the outcomes of this study, fire recurrences must be mitigated under the effects of anthropogenic climatic change to conserve plant biodiversity and soil functionalities and to avoid ecosystem collapse, especially when areas have already been burned already once.

## Figures and Tables

**Table 1 plants-12-01112-t001:** Soil physical, chemical and biological properties, their acronyms and the names of methods used for their quantifications.

Parameter	Code	Method Name and References
Bulk density (g.cm^−3^)	BD	common core method, mass and volume [46]
Granulometry (%)	sand, silt, clay	hydrometry [47]
Soil pH	pH	Standard, pH-H2O [48]
Lime content as the Total Neutralizing Value (Lime), as a percentage of Calcium Carbonate Equivalent (%CCE)	TNV	titration with NaOH [49]
Electrical conductivity (dS.m^−1^)	EC	aqueous soil extract [48]
Total nitrogen (%)	N	Kjeldahl method [50]
Available phosphorus (mg.kg^−1^, ppm)	P	spectrophotometer [51]
Soil organic matter (%)	SOM	chromic acid titration [52]
Microbial biomass carbon of soil (mgCO_2_.g^−1^)	MBC	[53]
Soil basal respiration (mgCO_2_.g^−1^.d^−1^)	SR	[54]
Substrate-induced respiration (mgCO_2_.g^−1^.d^−1^)	SIR	[55]
Soil urease enzymatic activity based on soil dry weight (dwt) and per hour (µgNH_4_^+.^g^−1^_soil dwt_.h^−1^)	Urease	colorimetry [56]

**Table 2 plants-12-01112-t002:** Community diversity indices used in the present study, ordered by increasing evenness, where *S* is the number of taxa *i* per sample, *N* is the total number of individuals observed per sample, and *P_i_* is the relative abundance of each taxon *i*.

Index Name	Formula	Range of Values	Properties
Menhinick	$R=\frac{S}{\sqrt{N}}$	*R* > 0	Richness controlling for the effect of observation size (rarefied richness)
Gini-Simpson	$\;D=1-\overset{S}{\underset{i=1}{\sum }}P_{i}^{2}$	0 ≤ *D* < 1	Diversity highlighting abundant taxa
Shannon	$H^{\prime }=-\overset{S}{\underset{i=1}{\sum }}P_{i}ln\left(P_{i}\right)$	0 < *H*′ ≤ ln(*S*)	Diversity highlighting low-abundance taxa
Pielou’s evenness	E=H′H′max	0 < *E* ≤ 1	Determines the equality of the community in individuals’ number per taxon, where *H*′*max* = ln(*S*)

**Table 3 plants-12-01112-t003:** Outputs of one-way ANOVA for comparison of soil physical, chemical, and biological properties, and community diversity indices of plants and AMF under fire treatment with different fire frequency (two fires occurred, one fire occurred, control site). BD: bulk density; TNV: total neutralizing value; EC: electrical conductivity; pH: soil acidity; N: total nitrogen; P: available phosphorus; MBC: microbial biomass carbon; SR: basal soil respiration; SIR: substrate-induced respiration; SOM: soil organic matter (greyed: significant *p*-values).

Soil Physics			Soil Chemistry			Soil Biology		
Property	*F*	*p*	Property	*F*	*p*	Property	*F*	*p*
BD	29.134	<0.001	TNV	1.818	0.192	MBC	6.029	0.010
Sand	0.779	0.475	pH	4.009	0.037	SR	9.294	0.002
Silt	0.292	0.751	EC	10.013	0.001	SIR	7.586	0.004
Clay	1.240	0.314	N	4.791	0.022	Urease	7.444	0.005
			P	1.378	0.279	SOM	4.710	0.024
Plant community diversity			AMF community diversity					
Index	*F*	*p*	Index	*F*	*p*			
Richness	6.26	0.009	Richness	2.234	0.138			
Simpson	6.846	0.007	Simpson	0.892	0.428			
Shannon	6.846	0.007	Shannon	5.984	0.011			
Evenness	0.357	0.705	Evenness	0.942	0.409			

## Data Availability

The data will be available by contacting authors.

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
