# Peer review of "Decrease in Soil Functionalities and Herbs’ Diversity, but Not That of Arbuscular Mycorrhizal Fungi, Linked to Short Fire Interval in Semi-Arid Oak Forest Ecosystem, West Iran"

_plants, 2023, doi:10.3390/plants12051112_

Round 1
Reviewer 1 Report
This is an interesting paper that merits publication. My main concerns are:
1. The english language expression is good, but occasionally is quite confusing and needs careful editing.
2. Information is not always fully provided - e.g.in the table captions not all abbreviations are defined and in Fig. 1 abbreviations are defined that are not used.
3. My main concern is the use of "repeated fires" to describe two fires in 10 years. The control plus one fir plus two fires is a good experimental set up, but you need to describe the results carefully. I don't think two fires is best described as "repeated".
4. I think the title should be "mycorrhizal" not "mycorrhiza".
Author Response
This is an interesting paper that merits publication.
Author response: we warmly thank the reviewer for his/her comment as well as his/her recommendations.
My main concerns are:
- The english language expression is good, but occasionally is quite confusing and needs careful editing.
Author response: Indeed. The present revised text has been now edited by a professional.
- Information is not always fully provided - e.g.in the table captions not all abbreviations are defined and in Fig. 1 abbreviations are defined that are not used.
Author response: it has been attentively revised.
- My main concern is the use of "repeated fires" to describe two fires in 10 years. The control plus one fir plus two fires is a good experimental set up, but you need to describe the results carefully. I don't think two fires is best described as "repeated".
Author response: ‘Repeated’ is a common term, but we deleted “repeated” and changed by “short fire interval”.
- I think the title should be "mycorrhizal" not "mycorrhiza".
Author response: Indeed. Fixed. Thanks so much.
Further, all figures have been attentively checked, fixed when there were some problems of units of acronyms.
Finally, 6 references have been added to the present version:
- Blake, G.R.; Hartge, K.H. Bulk density. In: Klute, A. (Ed) Methods of Soil Analysis. Part I. Physical and Mineralogical Properties. Agronomy Series. ASA, Madison, WI, 1986, pp.363-375. https://doi.org/10.2136/sssabookser5.1.2ed.c13
- Carcaillet, C.; Desponts, M.; Robin, V.; Bergeron, Y. Long-term steady-state dry boreal forest in the face of disturbance. Ecosystems 2020, 23, 1075-1092 DOI: doi.org/10.1007/s10021-019-00455-w
- Dove, N.C.; Hart, S.C. Fire reduces fungal species richness and in situ mycorrhizal colonization: a meta-analysis. Fire Ecol. 2017, 13, 37-65. doi: 10.4996/fireecology.130237746
- Owen, S.M.; Patterson, A.M.; Gehring, C.A.; Sieg, C.H.; Baggett, L.S.; Fulé, P.Z. Large, high-severity burn patches limit fungal recovery 13 years after wildfire in a ponderosa pine forest. Soil Biol. Biochem. 2019, 139, 107616, https://doi.org/10.1016/j.soilbio.2019.107616
- Rossetti, I.; Cogoni, D.; Calderisi, G.; Fenu, G. Short-Term Effects and Vegetation Response after a Megafire in a Mediterranean Area. Land, 2022, 11, 12, 2328.
- Taudière, A.; Bellanger. J.M.; Carcaillet, C.; Hugot, L.; Kjellberg, F.; Lecanda, A.; Lesne, A.; Moreau, P.A.; Scharmann, K.; Leidel, S.; Richard, F. Diversity of foliar endophytic ascomycetes in endemic Corsican pine forests. Fung. Ecol. 2018, 36, 128-140. doi 10.1016/j.funeco.2018.07.008
Reviewer 2 Report
I reviewed the paper of Mirzaei et al. and I found many interesting ideas and results but some adjustment is needed to improve this manuscript.
Firstly, one critical aspect that should be clarified concerns the focus of the study: the Authors indicate that semi-arid oak forest have been analyzed but, in the study, a spectrum of elements is analyzed but no one with a forest value; I suppose that one or two fires in the last 10 years could transform a forest formation into another vegetation type but, if so, then the unburned forest could not be used as a control.
The introduction section is messy with paragraphs that don't seem to follow a logical order; I suggest to follow this scheme: ecosystems, biological aspects (flora, fungi and microorganisms) and soil properties for example.
The title is not informative of this study as the aims are much broader and not limited to the vascular and fungal flora.
L16: ecosystem compartments sounds strange.
L40: also in Mediterranean climate regions (see for example https://doi.org/10.3390/land11122328)
L45: I suggest to delete “size”
L78: delete “semi-arid”
L80: delete “or on the mortality of mature trees (Enright et al., 2011)” (included in biodiversity).
LL99-100: what does it mean “apart from the growing season”?
L100: change 21 plots by study areas
L146: among factors
LL257-259: delete this sentence
LL324-327: the paper suggested above might help you implement this part of the discussion.
There are several grammatical and typo errors in the text.
Author Response
I reviewed the paper of Mirzaei et al. and I found many interesting ideas and results but some adjustment is needed to improve this manuscript.
Author response: we warmly thank the reviewer for his/her comment as well as his/her recommendations.
Firstly, one critical aspect that should be clarified concerns the focus of the study: the Authors indicate that semi-arid oak forest have been analyzed but, in the study, a spectrum of elements is analyzed but no one with a forest value; I suppose that one or two fires in the last 10 years could transform a forest formation into another vegetation type but, if so, then the unburned forest could not be used as a control.
Author response: That is an interesting comment of the reviewer. It is often considered that forest is the tree cover. But forest is much more than the tree-cover. Here, fires that occurred in this area during last years have not changed significantly the forest type. Although some oak trees were completely burned, many trees remained standing and alive following fire. Oak, as dominant species, has a high resilience ability based on resprouting and therefore, after fires, the oak cover re-occur in a short period of time. However, if the tree-cover does not change, it is not exact to consider that the forest has not changed. Forest is more than the simple tree-cover. The forest ecosystem also includes plants, fungi and other microorganisms, animals, and of course all soil physicochemical features. That is why the term forest is correct, even if tree-cover did not change. Based on our assumption, soil, fungi, and understory vegetation can be good proxies of ecosystem changes.
The introduction section is messy with paragraphs that don't seem to follow a logical order; I suggest to follow this scheme: ecosystems, biological aspects (flora, fungi and microorganisms) and soil properties for example.
Author response: Done. The second paragraph (soil) has been moved after the paragraph on microorganisms.
The title is not informative of this study as the aims are much broader and not limited to the vascular and fungal flora.
Author response: Correct. The title has been changed to also evoke soil functionalities that have been changed.
L16: ecosystem compartments sounds strange.
Author response: we changed it to ecosystem features. However, soil is an ecosystem compartment (cf. Odum and Odum, 1953, Fundamental of Ecology)
L40: also in Mediterranean climate regions (see for example https://doi.org/10.3390/land11122328)
Author response: Thanks. However The Zagros ecosystem belong not to the Mediterranean biome (https://en.wikipedia.org/wiki/Zagros_Mountains_forest_steppe https://en.wikipedia.org/wiki/Biome#/media/File:Vegetation.png) but to the warm temperate. It is the southern temperate ecosystem of the northern hemisphere. The reviewer could also have mentioned the tropical and the boreal ones. Therefore, we changed the sentence by adding a mention to Mediterranean, tropical and boreal biomes, and adding one reference.
L45: I suggest to delete “size”
Author response: change in fire sizes is an important source of ecosystem alteration notably by affecting the plant colonisation because the fire size hinder the seed dispersion by wind but also by zoochory or the soil fungal community (Ordonez et al. 2005; Doi: 10.1016/j.foreco.2004.10.067; Christopoulou et al. 2019 Doi : 10.1071/WF18153 ; Owen et al 2019 10.1016/j.soilbio.2019.107616). Therefore, we conserve ‘size’ in the sentence.
L78: delete “semi-arid”
Author response: Done.
L80: delete “or on the mortality of mature trees (Enright et al., 2011)” (included in biodiversity).
Author response: Done.
LL99-100: what does it mean “apart from the growing season”?
Author response: Modified.
L100: change 21 plots by study areas
Author response: Done.
L146: among factors
Author response: Done.
LL257-259: delete this sentence
Author response: Why not? Anyway, it has been done.
There are several grammatical and typo errors in the text.
Author response: The present revised text has been edited by a professional.
Further, all figures have been attentively checked, fixed when there were some problems of units of acronyms.
Finally, 6 references have been added to the present version:
- Blake, G.R.; Hartge, K.H. Bulk density. In: Klute, A. (Ed) Methods of Soil Analysis. Part I. Physical and Mineralogical Properties. Agronomy Series. ASA, Madison, WI, 1986, pp.363-375. https://doi.org/10.2136/sssabookser5.1.2ed.c13
- Carcaillet, C.; Desponts, M.; Robin, V.; Bergeron, Y. Long-term steady-state dry boreal forest in the face of disturbance. Ecosystems 2020, 23, 1075-1092 DOI: doi.org/10.1007/s10021-019-00455-w
- Dove, N.C.; Hart, S.C. Fire reduces fungal species richness and in situ mycorrhizal colonization: a meta-analysis. Fire Ecol. 2017, 13, 37-65. doi: 10.4996/fireecology.130237746
- Owen, S.M.; Patterson, A.M.; Gehring, C.A.; Sieg, C.H.; Baggett, L.S.; Fulé, P.Z. Large, high-severity burn patches limit fungal recovery 13 years after wildfire in a ponderosa pine forest. Soil Biol. Biochem. 2019, 139, 107616, https://doi.org/10.1016/j.soilbio.2019.107616
- Rossetti, I.; Cogoni, D.; Calderisi, G.; Fenu, G. Short-Term Effects and Vegetation Response after a Megafire in a Mediterranean Area. Land, 2022, 11, 12, 2328.
- Taudière, A.; Bellanger. J.M.; Carcaillet, C.; Hugot, L.; Kjellberg, F.; Lecanda, A.; Lesne, A.; Moreau, P.A.; Scharmann, K.; Leidel, S.; Richard, F. Diversity of foliar endophytic ascomycetes in endemic Corsican pine forests. Fung. Ecol. 2018, 36, 128-140. doi 10.1016/j.funeco.2018.07.008
Round 2
Reviewer 2 Report
I have found the new versionof the MS greatly improved and have no further requests for the authors.